# Effects of Decreasing Fishmeal as Main Source of Protein on Growth, Digestive Physiology, and Gut Microbiota of Olive Flounder (*Paralichthys olivaceus*)

**DOI:** 10.3390/ani12162043

**Published:** 2022-08-11

**Authors:** Bong-Seung Seo, Su-Jin Park, So-Yeon Hwang, Ye-In Lee, Seung-Han Lee, Sang-Woo Hur, Kyeong-Jun Lee, Taek-Jeong Nam, Jin-Woo Song, Jae-Sig Kim, Won-Je Jang, Youn-Hee Choi

**Affiliations:** 1Department of Fisheries Biology, Pukyong National University, Busan 48513, Korea; 2Aquafeed Research Center, National Institute of Fisheries Science, Pohang 37517, Korea; 3Department of Marine Life Science, Jeju National University, Jeju 63234, Korea; 4Future Fisheries Food Research Center, Institute of Fisheries Sciences, Pukyong National University, Busan 46041, Korea; 5Jeju Fish-Culture Fisheries Cooperatives, Jeju 63021, Korea; 6Department of Biotechnology, Pukyong National University, Busan 48513, Korea; 7Division of Fisheries Life Sciences, Pukyong National University, Busan 48513, Korea

**Keywords:** fishmeal, olive flounder, growth hormone, insulin-like growth factor 1, histology, digestive enzyme, microbiota

## Abstract

**Simple Summary:**

The demand for fishmeal is increasing due to aquaculture development, but the supply is unstable. This indicates the need to reduce the fishmeal content in the feed and develop an optimal fish-feed formulation through substitutes. However, most studies on reducing fishmeal content in feed were conducted at the laboratory level. In this study, the application of a low-fishmeal diet as feed to olive flounder was evaluated in terms of growth-related factors, digestive physiology, and microbiota raised for five months in a fish farm using four feed formulations- FM70 [control (CON), 70% fishmeal], FM45 (45% fishmeal), FM35A (35% fishmeal), and FM35B (35% fishmeal + insect meal). There was no difference in growth-related factors, digestive physiology, and gut microbiota diversity compared with the CON-fed fish. Therefore, reducing the fishmeal content of the feed by up to 35% does not adversely affect growth and physiological characteristics under farm conditions.

**Abstract:**

In olive flounder (*Paralichthys olivaceus*), growth performance, expression of growth-related factors, digestive physiology, and gut microbiota were assessed under farm conditions in the fish fed diets with low levels of fishmeal. Four experimental diets were prepared, FM70 [control (CON), 70% fishmeal], FM45 (45% fishmeal), FM35A (35% fishmeal), and FM35B (35% fishmeal + insect meal), and fed to the fish for five months. The CON-fed fish had the highest plasma GH, but IGF-1 and hepatic IGF-1 mRNA expression of the olive flounder fed diets with low-fishmeal levels did not significantly differ among diets. The intestinal villus length, muscular thickness, and the number of goblet cells were statistically similar, and ocular examination of hepatopancreas showed no discernable difference in all experimental diets. The chymotrypsin content of FM35B-fed fish is significantly lower, but trypsin and lipase contents are similar. The diversity of gut microbiota did not differ among groups, although the FM35B group had a higher composition of Firmicutes. Thus, a diet with reduced fishmeal content and several alternative protein sources can be used as feed ingredients in feed formulation for olive flounder reared under typical aquaculture farm conditions.

## 1. Introduction

A commercially significant aquaculture fish species, the olive flounder (*Paralichthys olivaceus*), contributed 46.7% of Korea’s total aquaculture output in 2021 [1]. During aquaculture production, fish are fed well-balanced formulated diets to supply ample nutrients necessary for growth and development. Protein, one of the components in fish-formulated diets, is crucial for delivering nutritional value and is mainly obtained from fishmeal. Because olive flounder is a carnivorous fish with a high protein need, its total feed must contain up to 60% fishmeal content [2]. However, the unstable supply of fishmeal in the market caused by the decline in fish catches, environmental pollution, and overfishing is expected to unmeet the demand of the rapidly growing aquaculture industry [3]. Thus, the development of additional protein-sourced feed that can stably replace fishmeal is now seen as a necessity for fish aquaculture [4]. Therefore, various protein replacement studies have been conducted to evaluate the potential of emerging alternative ingredients in olive flounder feed formulation. These include the utilization of animal and plant-based proteins from poultry- and tuna-by-products, insect meal, SBM, and SPC, and the proper mixing of these ingredients would increase the replacement rate of fishmeal in feed [5,6,7,8,9]. In addition, in recent research, black soldier fly (*Hermetia illucens*) with amino acid composition similar to fishmeal is being used as a substitute for fishmeal [10]. Most studies have been conducted in closed, controllable, and stable environments (such as laboratory conditions). However, it is very important to know the effect of feed ingredients on fish under changed environmental conditions because the conditions are different from that of the farm, which is an open environment [11]. 

It is vital to create diets for various fish weights since, in general, the nutritional needs of fish differ as they grow [12]. The growth-related factors, digestive enzyme activity, tissue development, and gut microbiota may vary depending on protein sources in the diet [13,14,15,16]. The teleost growth hormone (GH) acts on the liver to produce more than 70% of insulin-like growth factor 1 (IGF-1) and promote secretion [17]. IGF-1 is transported through the blood to target organs such as bones and muscles, where it promotes protein synthesis or cell division and is used as a growth indicator in regulating growth and nutrient metabolism [18]. As the fish grow, the nutrients in the diet influence the development of intestinal villus and muscular thickness [19]. In addition, the digestive enzyme activity secreted from the intestine and the composition of gut microbiota is closely related to fish’s growth and health and are used as indicators of feed digestion and nutritional status [20,21]. Therefore, it is essential to simultaneously look into the influence on growth and digestive physiology to choose the best mixing ratio of fish fed with the low-fishmeal formulation. So far, studies have been conducted on growth factors, digestive enzyme activity, tissue development, and gut microbiota of olive flounder [22,23,24]. However, the number of studies on the low-fishmeal formulations proposed by synthesizing these factors are few, and the content is insufficient.

Our research team has been researching to find the appropriate ratio of fishmeal substitutes and additives from larvae (less than 10 g) to growing olive flounder (less than 300 g) raised on a farm where olive flounder production is taking place [23,25]. However, it is also important to study the effect of various mixing ratios of feed ingredients in a long-term breeding experiment from a 300 g or more sub-adult fish to a commercial-size (about 1 kg) fish in actual aquaculture conditions. The current study employed open farms where environmental variables constantly change, in contrast to most research conducted in closed, stable environments (i.e., research institute). It is then crucial to understand the impact of feed ingredients on fish under changing environmental conditions [11]. Hence, this study was conducted to evaluate the appropriateness of the formulated feed prepared by the improved mixing ratio by examining growth-related factors, digestive physiology, and diversity and composition of gut microbiota of commercial-size olive flounder (about 1 kg) reared under typical aquaculture farm conditions.

## 2. Materials and Methods

### 2.1. Experimental Diets

The feed formulation and proximate composition of the diets used in this study are shown in Table 1. The diets are improved formulations previously designed by Jo et al. [23]. Four diet treatments were prepared with various fishmeal contents: 70% fishmeal [FM70, control (CON)], 45% fishmeal (35% fishmeal replacement, FM45), 35% fishmeal (50% fishmeal replacement, FM35A), and 35% fishmeal with the addition of 3.5% insect meal and 0.5% insect oil (50% fishmeal replacement, FM35B). Insect meal and insect oil were from the black soldier fly larvae (BSFL). Betaine and taurine were added to improve the taste of the diets with low-fishmeal levels. In addition, methionine was added to prevent methionine deficiency due to fishmeal replacement. Fish oil and calcium monophosphate produced the same crude lipid and phosphorus levels in each experimental feed. Fishmeal and BSFL were supplied by Lota Protein S.A., Chile and C.I.E.F. Company, Korea. Experimental feeds were prepared following dietary formulations using a feed maker (Jiangsu Muyang Corp. Ltd., Yangzhou, China) and oven-dried at 85 °C. The feeds were stored at −20 °C before use.

### 2.2. Experimental Fish and Sampling

The experiment was conducted at Shenyang Fisheries in Seongsan-eup, Seogwipo-si, Jeju, Republic of Korea. Fish were kept in a mixture of natural seawater and groundwater (7:3 ratio) in 10 m^2^ concrete tanks. Around 2800 olive flounders with an initial average weight of 365 ± 60 g per tank were raised in a total of 12 tanks, which represents four treatments replicated thrice. The experiment was conducted from April to September 2021. Fish were hand-fed twice daily at 07:30 and 16:00 until apparent satiation levels, and the rearing conditions were temperature 16.9–17.8 °C, salinity 29.5–31.2 ppt, pH 6.9–7.65, DO 7.01–7.61 mg/L, and was measured once every morning. 

Fish sampling procedures were conducted as follows. Before sampling, fish were starved for 24 h. In each experimental group, 20 individuals were collected at random and anaesthetized with 2-phenoxyethanol (100 ppm). The total weight (g), total length (mm), body height (mm), and body width (mm) of fish were then measured to facilitate calculations of weight gain (WG), specific growth rate (SGR), and condition factor (CF). The liver and digestive tract were extracted and weighed (g) to facilitate calculations of the hepatosomatic index (HSI) and viscerosomatic index (VSI). Blood was collected from the caudal vasculature using a syringe (Jungrim, Seoul, Korea) treated with a 0.5 M ethylenediaminetetraacetic acid (EDTA) solution to prevent coagulation. Thereafter, blood samples were transferred to 1.5 mL tubes treated with EDTA solution and centrifuged at 5000 rpm and 4 °C for 15 min. Liver samples were removed and cut into two portions, one for hepatic IGF mRNA analysis and one for histological examination. For digestive enzyme analysis, intestinal samples were snap frozen in liquid nitrogen. The intestine and liver were fixed in 10% formalin solution and used for histological analyses. Except for samples to be used for histological examinations, all samples were stored at −70 °C until analysis.

### 2.3. Quantitative Analysis of Plasma GH and IGF-1

Plasma GH and IGF-1 concentrations were analyzed using a Fish Enzyme-linked Immunosorbent Analysis (ELISA) Kit (CUSABIO, Houston, TX, USA) following the manufacturer’s instructions. Specifically, 50 µL plasma samples were used, and absorbance was read at 450 nm in an EZ Read 400 microplate reader (Biochrome, Cambridge, UK) and calculated as pg/mL units based on standard curve analysis.

### 2.4. IGF-1 Gene Transcript Analysis

Liver samples were thawed, cut into small pieces, homogenized using RNAiso (Takara, Kusatsu, Japan), and allowed to react sufficiently at 4 °C for 1 h. Subsequently, 200 µL of chloroform was added and allowed to react for 10 min before centrifugation at 12,000 rpm at 4 °C for 10 min to obtain the supernatant.

The supernatant was reacted with isopropyl alcohol and centrifuged under the abovementioned conditions to produce RNA pellets. The separated RNA pellets were then washed with 75% and 100% ethanol, dissolved in PCR water, and quantitatively analyzed for total RNA using a UV/VIS Nano spectrophotometer (Micro Digital, Seongnam, Korea). Subsequently, cDNA was synthesized using a PrimeScript First Strand cDNA Synthesis Kit (Takara, Kusatsu, Japan) according to the manufacturer’s method, and the PCR mixture was obtained using EmeraldAMP GT PCR Master Mix (Takara, Kusatsu, Japan) was electrophoresed on 2% agarose gel. The primers used in the analysis were designed based on the base sequences shown in Table 2, and the expression of each gene was analyzed using relative quantification of the target DNA (IGF-1) and a housekeeping gene (18s ribosomal RNA).

### 2.5. Histological Analysis

The intestine and liver tissues were subjected to histological analysis following procedures employed by Jo et al. [23]. Briefly, tissues were dehydrated using increasing alcohol concentrations and cleaned using xylene solutions. The tissues were then embedded in paraffin blocks sectioned at 5–7 µm. Intestine and liver tissues were stained with hematoxylin (Dako, Carpinteria, CA, USA) and eosin (Sigma, St. Louis, MO, USA) (H&E) and Alcian blue/periodic acid (Sigma Aldrich, St. Louis, MO, USA) and Schiff stain (Merck, Darmstadt, Germany) (AB-PAS) (pH 2.5) were used to stain the separate intestine samples. The stained tissues were mounted using Canada balsam (Junsei, Tokyo, Japan). All histological slides were observed using an OLYMPUS SEX41 microscope (Olympus, Tokyo, Japan). Micrographs, villus length, and goblet cell counts were analyzed using MIchrome 6 (Tucsen, Fuzhou, China) and Mosaic 2.1 (Tucsen, Fuzhou, China). 

### 2.6. Digestive Enzyme Analysis

Intestine samples were homogenized in 1 mL of PBS and centrifuged at 5000 rpm at 4 °C for five min. The collected supernatant was used to analyze trypsin, chymotrypsin, and lipoprotein lipase using a Fish ELISA Kit (CUSABIO, Houston, TX, USA) according to the manufacturer’s instructions. Supernatant samples were used and absorbance was read as described in Section 2.3.

### 2.7. Gut Microbiota Analysis

Total DNA of the gut microbiota was isolated using a FavorPrep Tissue Genomic DNA Extraction Mini Kit (Favorgen Biotech Corp., Pingtung, Taiwan). The V3-V4 region of 16s rRNA was amplified using the forward and reverse primers 5′-TCGTC GGCAG CGTCA GATGT GTATA AGAGA CAGCC TACGG GNGGC WGCAG-3′ and 5′-GTCTC GTGGG CTCGG AGATG TGTAT AAGAG ACAGG ACTAC HVGGG TATCT AATCC-3′, respectively. An Illumina HiSeq platform was used for sequencing. Operational taxonomic units with 97% similarity were used for gut microbiota composition analysis at the phylum and genus levels. Alpha diversity was estimated based on the ACE, Chao1, Jackknife, Shannon, and Simpson indexes. Beta diversity analysis included using principal coordinates analysis based on UniFrac metrics to assess differences between groups [27,28,29,30,31].

### 2.8. Statistical Analysis

Data obtained using ELISA and reverse transcription polymerase chain reaction are shown as means ± standard error of the mean and were analyzed using one-way ANOVA via SPSS Statistics version 27 (IBM, New York, NY, USA). If a significant difference was detected via ANOVA, Duncan’s post-hoc test was run to determine a significant difference among treatment means at *p* < 0.05. 

## 3. Results

### 3.1. Growth Performance

The growth performance of olive flounder fed experimental diets for five months is shown in Table 3. There were no significant differences in FW, WG, SGR, CF, HSI, VSI, and survival in all experimental diets (*p* > 0.05).

### 3.2. Growth-Related Factors

Plasma GH activity was significantly higher in the CON-fed olive flounder compared to other experimental diets (Figure 1a, *p* < 0.05). In contrast, hepatic IGF-1 mRNA expression and plasma IGF-1 activity did not significantly differ among diets (Figure 1b,c, *p* > 0.05). 

### 3.3. Histological Analysis of the Intestine and Liver

The histological structure of the intestine is shown in Figure 2. The intestinal villus length, muscular thickness, and the number of goblet cells were not significantly different among the experimental groups (Table 4, *p* < 0.05).

The histological structure of the liver is shown in Figure 3. The ocular inspection of the hepatic tissues showed that hepatocyte organization, morphology, and pancreatic development were similar in all treatment groups. In addition, no steatosis and vacuolization due to fat accumulation were observed. The zymogen granules found in the acinar cells of the exocrine pancreatic islet were observed to be similar in all experimental groups (Figure 3). 

### 3.4. Digestive Enzyme Activity

The digestive enzyme (trypsin, chymotrypsin, and lipase) activity of olive flounder is shown in Table 5. Trypsin and lipase activity did not statistically differ among experimental diets (*p* > 0.05). However, the chymotrypsin activity of FM35B-fed fish is significantly lower compared to other diets tested (*p* < 0.05). 

### 3.5. Diversity of Gut Microbiota

At the end of the experimental period, the total DNA of olive flounder intestinal bacteria was isolated, and microbiota analysis was performed based on the V3–V4 region of the 16s rRNA sequence. Table 6 shows the results of the α-diversity of olive flounder intestinal bacteria according to the experimental diet group. The richness estimators, ACE, CHAO, and Jackknife, as well as the diversity estimators, Shannon and Simpson, were not significantly different among the experimental diet groups (*p* > 0.05). Similarly, the *β*-diversity analysis did not reveal significant differences among these groups (Figure 4).

### 3.6. Gut Microbiota Composition

The most abundant phylum in all diet groups was Proteobacteria. Unlike other groups, including CON, the FM35B group contained a high proportion of Firmicutes. At the genus level, *Ralstonia* and *Sphingomonas* were abundant in all groups. Notably, *Vibrio* was most abundant in the CON group (Figure 5).

## 4. Discussion

In general, highly nutritious feed enhances fish growth, although the nutrients and proportions required to depend on the fish species. Kim et al. [8] reported that replacing fishmeal with acid-concentrated soybean meal could reduce the fishmeal content of feed by up to 40%, but the growth rate of olive flounder decreased when fishmeal was reduced to ≤35%. This is due to the limited availability of amino acid content in the feed when fishmeal is replaced with a single ingredient and the anti-nutritional effect of plant ingredients. Fishmeal contains nutrients essential for fish growth and is difficult to replace with just a single ingredient. In order to supplement the nutrients needed for growth, it is, therefore, crucial to understand the composition and ratio of various alternative ingredients. In olive flounder, Kim et al. [32] reported that it was possible to reduce fishmeal content by up to 32.5% by properly mixing animal and plant ingredients, such as wheat gluten, soy protein concentrate, greaves, and poultry by-products. The present study showed no significant difference in FW, WG, SGR, CF, HSI, and VSI in olive flounder fed experimental diets. The formulated feed with reduced fishmeal content by up to 35% did not adversely affect the growth of juvenile olive flounder [23], which is consistent with the results of the current study. Furthermore, it was reported that mixing several alternative ingredients can supplement inadequate nutrients and replace high proportions of fishmeal [33]. Therefore, due to insignificant variation in the growth performance of olive flounder, it is inferred that formulated diet with reduced fishmeal content of up to 35% with dietary supplementation of several alternative protein sources can be used as feed up to commercial size.

GH is mainly regulated by the content of IGF-1 released into the blood through phosphorylated hepatic IGF-1 mRNA and the binding of GH–GH receptors [34]. IGF-1 circulates in the blood and acts on tissues to promote cell differentiation, skeletal formation, and muscle synthesis and inhibit GH secretion from the pituitary gland through a feedback process [35]. Fish growth occurs through the GH–IGF axis, and some studies have reported that the activity of this axis is influenced by nutrient sources in the diet [36]. When barramundi (*Lates calcarifer*) was fed with feed containing different crude protein contents for six weeks, blood IGF-1 was positively correlated with protein content [37]. Similar results were found in studies of other fish species, including olive flounder, Asian red-tailed catfish (*Hemibagrus wyckioides*), gilthead sea bream (*Sparus aurata*), and olive rockfish (*Sebastes serranoides*) [22,26,38,39]. Thus, IGF-1 provides information about the optimal protein level required for various fish species [35]. In the present study, plasma GH activity was significantly lower in olive flounder fed with reduced fishmeal content than CON. However, hepatic IGF-1 mRNA expression level and plasma IGF-1 activity were not significantly different in all experimental groups. GH stimulates the expression of IGF-1 mRNA produced in the liver, ultimately regulating the IGF-1 activity in the blood. However, in the present study, it might not substantially stimulate the IGF-1 expression in the liver. This was also observed in common carp (*Cyprinus carpio*), wherein higher GH expression did not significantly influence plasma IGF-1 content in 50% or 75% FM-replaced SBM diet [40]. Even with the decreased GH activity in a reduced fishmeal diet, the observed insignificant difference in hepatic IGF-1 mRNA expression level is probably because the liver was stimulated substantially through the supply of nutrients in all experimental groups. In addition, the similar activity of IGF-1 in the blood suggests that the protein requirement was sufficiently satisfied even when the fishmeal content in the feed was reduced by 35%.

The intestine works to absorb and digest food and is divided into the front, middle, and posterior parts based on histological characteristics. It comprises the mucous membrane, muscular layer, and serous membrane. The villus increases the intestine’s surface area, which improves nutrient absorption [41].

On the other hand, the olive flounder liver is associated with the pancreas and plays an important role in growth-related endocrine physiology, including the secretion and metabolism of digestive fluid. Therefore, analyzing the development of these tissues using histological techniques can provide crucial information on the digestion and absorption of feed [42]. 

In largemouth bass (*Micropterus salmoides*), intestinal villi length and muscle layer thickness were reduced in an experimental group fed feed with a low-fishmeal content, reportedly due to the lack of peptides and other active substances in the feed [43]. The hybrid grouper (*Epinephelus fuscoguttatus* ♀ × *Epinephelus lanceolatus* ♂) fed with fishmeal replaced by up to 70% with PBM did not differ in the gut micromorphology of the fish. In another trial, fish fed reduced fishmeal with increasing concentration of SBM and PBM ratio produced inferior gut morphology such as fold height, enterocyte height, and microvillus height [44]. In this study, the development of intestinal villus length and muscular thickness was similar in all experimental diets, confirming that no histological changes were related to reduced fishmeal content and utilization of other feed ingredients. The unbalanced nutritional availability in the diet can result in several hepatic histological abnormalities such as hepatocyte swelling and steatosis. 

In the study by Zhou et al. [44], in hybrid grouper, FM replaced by 50% PBM or higher resulted in increasing occasions of steatosis in the hepatocytes compared to low PBM replacement ratios. In the same research study, the increased amount of SBM in the diet produces an observable swelling of the hepatocytes. In the current study, lipid accumulation and hepatocyte swelling in the liver tissues were not observed in all experimental diets. Taken together, the low level of fishmeal in combination with various animal and plant proteins did not negatively affect intestinal and hepatic tissue composition and structure. The goblet cells in the mucosal epithelium secrete mucus to aid lubrication, promoting absorption by increasing intestinal motility and digestive activity. In addition, the number of goblet cells increases with the proper proportion of nutrients in feed, which can increase the growth and immunity of organisms; thus, goblet cell count can be used as an indicator of gut health [14,45]. 

In a previous study on olive flounder, goblet cell count decreased relative to that in the control group when the fishmeal content was replaced with a single ingredient; however, there was no significant difference in an experimental group fed feed to which the hydrolysate of shrimp and hydrolysate of tilapia were added [46]. In the current study, the number of goblet cells in the intestine was similar in all experimental diet groups, suggesting that feeding with low-fishmeal content did not negatively affect the number of goblet cells and could positively affect intestinal absorption and protection due to increased mucus secretion.

Digestive enzyme activity can be controlled by the nutrient source of the feed and potentially affects growth. [16,47]. In juvenile olive flounder, Jo et al. [23] reported that reduced fishmeal in the diet did not influence trypsin, chymotrypsin, and lipase activity. However, in the current study, digestive enzyme activity except for chymotrypsin was similar among experimental groups, likely because reduced fishmeal in the feed was replaced with vegetable and animal protein appropriately and substantially. The reduced activity of chymotrypsin in FM35B-fed fish is likely due to two major factors known to affect chymotrypsin activity: internal factors (e.g., trypsin phenotype and fish life-stage) and external factors (e.g., water temperature, starvation, feeding, and nutritional condition) [48]. The lower chymotrypsin in FM35B-fed fish is unlikely due to internal factors, which is the same in all treatment groups. The low level of fishmeal and the addition of mixed animal and plant protein might influence the chymotrypsin activity. However, further research is required to elucidate its mechanism of action.

The gut microbiome of fish is altered by various factors, including habitat, growth stage, and feeding activity [49,50,51]. An imbalance in the gut microbiota can lead to common metabolic conditions, such as fatty liver, oxidative stress, and inflammatory responses [52,53]. These metabolic changes may negatively affect the growth, development, and health of fish [52,53]. Therefore, many studies have investigated the maintenance and/or regulation of the intestinal microbiota balance to increase fish growth and immunity. In the present study, there was no significant difference in α-diversity among experimental diet groups, and no clear differences were identified among groups in the β-diversity analysis. 

According to Boland et al. [54], lower gut microbial diversity can lead to physiological dysfunction, which in turn can lead to disease. The fishmeal replacement diet used in the current study did not seem to cause an imbalance in the intestinal microbiota of olive flounder, nor did it induce negative effects such as metabolic disease and physiological dysfunction. An analysis of microbial composition at the phylum level revealed that the dominant phylum in all groups was Proteobacteria, which is consistent with the previous analysis of healthy olive flounder [55,56]. However, Firmicutes were abundant only in the FM35B group. 

According to Niu et al. [57], the proportion of Firmicutes in the intestinal microbial community of farmed fish is strongly influenced by diet. Therefore, black soldier fly meal and black soldier fly oil, which were present only in the FM35B diet, may have affected the increase in Firmicutes. Further studies are required to determine the effect of increasing the gut microbiota levels of Firmicutes on digestibility and growth in olive flounder. Analysis of microbial composition at the genus level showed that *Vibrio* was abundant only in the CON group. *Vibrio* is a pathogenic strain that causes vibriosis, which in turn causes persistent damage in olive flounder aquaculture [58]. In the present study, reducing fishmeal content in feed apparently reduced the proportion of *Vibrio* in the intestinal microbiota of olive flounder, which could be an important result for disease control studies in olive flounder aquaculture.

The incorporation of BSFL meal in feed formulation has recently gained attention in nutrition studies due to its high protein content, waste-recycling capacity, and sustainability. However, the addition of BSFL meal in the FM35B diet did not produce a positive growth effect on olive flounder compared to other diets. In Atlantic salmon (*Salmon salar*), a diet supplemented with 5–25% BSF meal with amino acids (Lys and Met) had a positive effect on fish performance [59]. In barramundi (*Lates calcarifer*), when PBM replaced 45% over fishmeal supplemented with 10% BSFL showed a positive effect on growth and histology. However, when 90% of FM was replaced, growth was negatively affected [60]. Further, when BSF meal and tuna hydrolysate were supplemented at 5% each, growth performance improved even when FM was completely replaced with PBM [61]. These would suggest effect varies based on the utilization percentage of BSF and other protein source ingredients in feed formulation in a range of cultured fish species. In juvenile olive flounder, Jo et al. [23] reported that a diet supplemented with 7% BSFL did not significantly differ from the basal diet, although FW and WG values were the lowest among diets. In this study, compared to FM35A, results suggest no apparent efficacy of the BSFL supplementation in the diet, given that FW and WG showed the lowest values despite supplementation with amino acids and several protein sources. This suggests that the BSFL meal given has low food value for olive flounder even at a low supplementation ratio (3.5%).

## 5. Conclusions

The growth, levels of growth-related factors, digestive physiology, and intestinal microbiota composition were similar in all experimental diets tested in this study. Thus, a diet formulated with reduced fishmeal content by up to 35% and incorporating various animal and plant proteins can be used as feed for commercial-size olive flounder reared under farm conditions. Further, it is considered that this study’s low-fishmeal-feed mixture ratio can be used from larvae to commercial-size fish.

## Figures and Tables

**Figure 1 animals-12-02043-f001:**
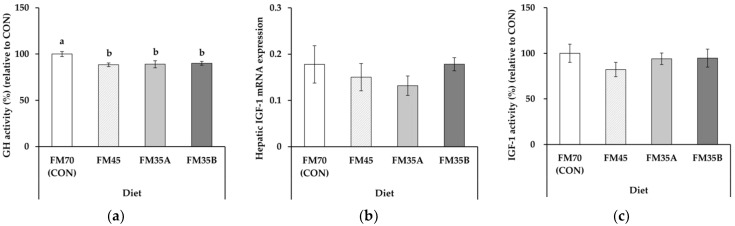
Growth-related factors of olive flounder (Paralichthys olivaceus) fed experimental diets for five months (*n* = 10). (**a**) Plasma GH activity; (**b**) hepatic IGF-1 mRNA expression; (**c**) plasma IGF-1 activity. FM70 (CON), 70% fishmeal diet; FM45, 45% fishmeal diet; FM35A, 35% fishmeal diet; FM35B, 35% fishmeal + 3.5% insect meal + 0.5% insect oil diet. Different letters above the bars indicate significant differences at *p* < 0.05.

**Figure 2 animals-12-02043-f002:**
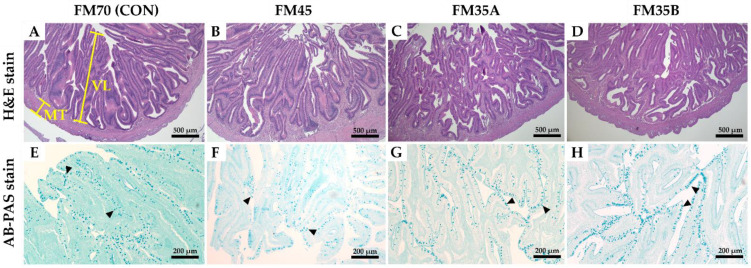
Histological structure of the intestine of olive flounder (*Paralichthys olivaceus*) fed experimental diets for five months. VL, villus length; MT, muscular thickness; goblet cell (black arrowhead) are shown. Representative micrographs of H&E stain ((**A**–**D**), image magnification, ×40 (scale bar = 500 µm)), AB-PAS stan ((**E**–**H**), image magnification, ×100 (scale bar = 200 µm)).

**Figure 3 animals-12-02043-f003:**
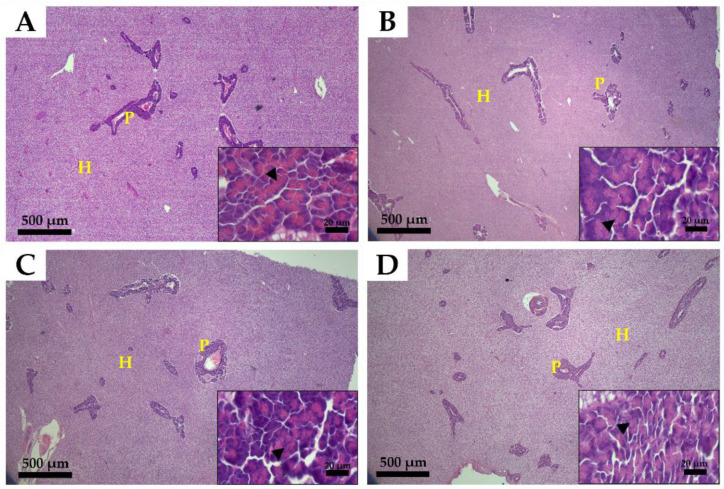
Histological structure of the liver of olive flounder (*Paralichthys olivaceus*) fed experimental diets for five months. (**A**), 70% fishmeal diet; (**B**), 45% fishmeal diet; (**C**), 35% fishmeal diet; (**D**), 35% fishmeal + 3.5% insect meal + 0.5% insect oil diet. P, pancreas; H, hepatocyte; zymogen granules (black arrowhead) are shown. Main image magnification, ×40 (scale bar = 500 µm); inset image magnification, ×1000 (scale bar = 20 µm).

**Figure 4 animals-12-02043-f004:**
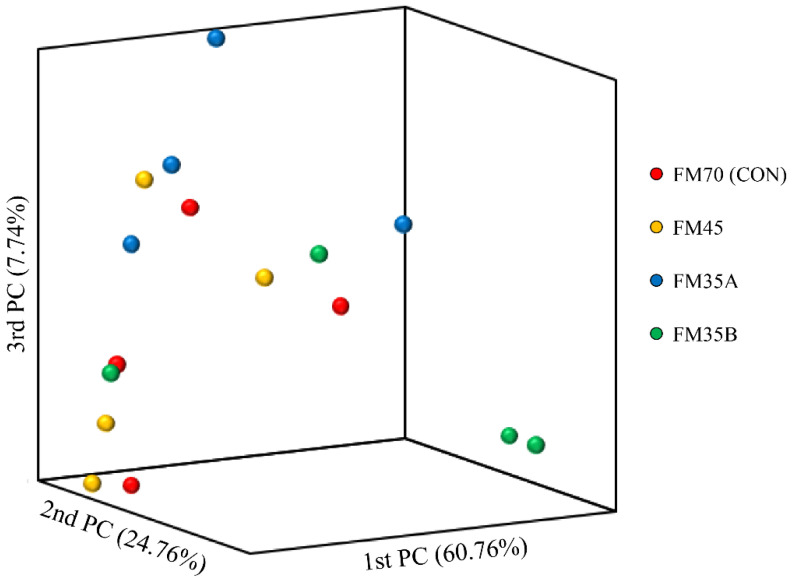
Principal coordinate analysis based on the weighted UniFrac metrics of bacterial operational taxonomic units among the different experimental diets.

**Figure 5 animals-12-02043-f005:**
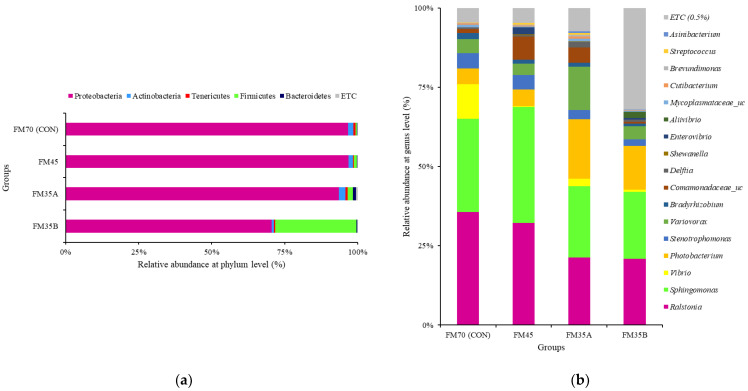
Relative abundance of the gut microbiota communities of olive flounder (*Paralichthys olivaceus*) fed different experimental diets. Analysis at the phylum (**a**) and genus (**b**) levels.

**Table 1 animals-12-02043-t001:** Formulation and proximate composition of the experimental diets.

Ingredients (%)	Diet			
	FM70 (CON)	FM45	FM35A	FM35B
Sardine FM	35.00	22.50	17.50	17.50
Anchovy FM	35.00	22.50	17.50	17.50
Tankage meal	-	8.00	11.50	10.50
PBM ^1^	-	4.50	6.50	6.00
TBM ^2^	-	-	1.00	1.60
Wheat gluten	-	5.50	4.70	4.70
SPC ^3^	-	5.50	8.00	6.50
Starch	11.00	10.80	10.73	10.33
Soybean meal	12.00	12.00	12.00	12.50
BSFL ^4^	-	-	-	3.50
Fish oil	3.30	4.30	4.20	2.50
BO ^5^	-	-	-	0.50
Lecithin	0.50	0.50	0.70	0.70
Betaine	-	1.00	1.20	1.20
Taurine	-	0.50	0.80	0.80
Met 99%	-	-	0.07	0.07
MCP ^6^	0.50	0.70	0.70	0.70
Mineral mix	1.00	1.00	1.00	1.00
Vitamin mix	1.00	1.00	1.00	1.00
Vitamin C	0.10	0.10	0.10	0.10
Vitamin E	0.10	0.10	0.10	0.10
Choline	0.50	0.50	0.70	0.70
Proximate analysis (%, dry matter)				
Moisture	6.22	8.35	6.64	6.10
Crude protein	56.90	56.80	57.00	56.30
Crude lipid	7.55	9.17	8.04	8.86
Crude ash	14.60	12.70	12.30	14.00

^1^ Poultry by-product meal; ^2^ Tuna by-product meal; ^3^ Soy protein concentrate; ^4^ Black soldier fly larvae meal; ^5^ Black soldier fly oil; ^6^ Mono-calcium phosphate.

**Table 2 animals-12-02043-t002:** Oligonucleotide primer sequences used in reverse transcription polymerase chain reaction.

Primer Name		Sequence (5′-3′)	Amplicon Size (bp)	Genbank No.	Ref.
18s rRNA	Forward	GGTCTGTGATGCCCTTAGATGTC	107	EF126037.1	[25]
	Reverse	AGTGGGGTTCAGCGGGTTAC			
IGF-1	Forward	CGGCGCCTGGAGATGTACTG	144	AF016922.2	[26]
	Reverse	TGTCCTACGCTGTGCCT			

**Table 3 animals-12-02043-t003:** Growth performance of olive flounder (*Paralichthys olivaceus*) fed experimental diets for five months.

Growth Performance	Diet			
	FM70 (CON)	FM45	FM35A	FM35B
FW (g) ^1^	1045.48 ± 200.77	1069.81 ± 156.08	1012.05 ± 210.21	980.90 ± 142.37
WG (%) ^2^	192.81 ± 80.50	196.25 ± 77.84	181.10 ± 91.43	178.18 ± 83.92
SGR (%) ^3^	0.67 ± 0.17	0.68 ± 0.14	0.64 ± 0.20	0.64 ± 0.17
CF ^4^	2.31 ± 0.36	2.30 ± 0.25	2.30 ± 0.60	2.15 ± 0.23
HSI (%) ^5^	1.90 ± 0.36	1.72 ± 0.46	1.66 ± 0.23	1.88 ± 0.37
VSI (%) ^6^	4.18 ± 0.70	4.15 ± 0.49	4.02 ± 0.32	4.19 ± 0.56
Survival (%)	71.7 ± 0.04	70.7 ± 2.25	71.8 ± 1.79	73.7 ± 0.31

Values are shown as mean ± standard deviation (*n* = 20). The lack of superscript letters indicates no significant differences among treatments (*p* > 0.05). ^1^ Final weight (g). ^2^ Weight gain (%) = [(final mean body weight − initial mean body weight)/initial mean body weight] × 100. ^3^ Specific growth rate (%) = [(log_e_ final body weight − loge initial body weight)/days] × 100. ^4^ Condition factor = (final body weight/total length) × 100. ^5^ Hepatosomatic index (%) = (liver weight/final body weight) × 100. ^6^ Viscerosomatic index (%) = (visceral weight/final body weight) × 100.

**Table 4 animals-12-02043-t004:** Villus length, muscular thickness, and goblet cell count in the intestine of olive flounder (*Paralichthys olivaceus)* fed experimental diets for five months.

	Diet			
	FM70 (CON)	FM45	FM35A	FM35B
Villus length (μm)	1716.69 ± 156.66	1673.29 ± 127.24	1701.56 ± 149.23	1689.70 ± 102.67
Muscular thickness (μm)	224.06 ± 25.43	228.00 ± 27.37	222.88 ± 30.59	232.57 ± 19.88
Goblet cell (cells/1000 µm^2^)	358.60 ± 29.07	352.80 ± 28.44	365.00 ± 22.55	339.8 ± 15.35

Values are shown as mean ± standard deviation (*n* = 10). The lack of superscript letters indicates no significant differences among treatments (*p* > 0.05).

**Table 5 animals-12-02043-t005:** Digestive enzyme activity (trypsin, chymotrypsin, and lipase) of olive flounder (*Paralichthys olivaceus*) fed experimental diets for five months.

Enzyme (pg/mL)	Diet			
	FM70 (CON)	FM45	FM35A	FM35B
Trypsin	92.74 ± 2.45	91.13 ± 3.73	89.59 ± 2.27	87.23 ± 1.94
Chymotrypsin	50.32 ± 1.49 ^a^	50.72 ± 0.88 ^a^	50.19 ± 1.44 ^a^	45.56 ± 2.17 ^b^
Lipase	29.66 ± 1.40	29.76 ± 0.89	30.93 ± 0.55	31.10 ± 0.64

Values are the means ± standard error of the mean (*n* = 10). Different letters above bars indicate significant differences at *p* < 0.05.

**Table 6 animals-12-02043-t006:** Alpha diversity of the intestinal bacterial communities of olive flounder (*Paralichthys olivaceus*).

	Diet			
	FM70 (CON)	FM45	FM35A	FM35B
ACE	207 ± 24	228 ± 43	270 ± 49	269 ± 92
CHAO	199 ± 23	222 ± 41	263 ± 46	262 ± 90
Jackknife	213 ± 25	240 ± 45	281 ± 51	282 ± 99
Shannon	1.82 ± 0.07	1.94 ± 0.41	2.24 ± 0.51	1.89 ± 0.31
Simpson	0.28 ± 0.02	0.26 ± 0.09	0.24 ± 0.13	0.27 ± 0.06

Values are means ± standard deviation (*n* = 4). The lack of superscript letters indicates no significant differences among treatment groups (*p* > 0.05).

## Data Availability

Data are available upon reasonable request to the corresponding author.

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
