# Peer review of "Effects of Decreasing Fishmeal as Main Source of Protein on Growth, Digestive Physiology, and Gut Microbiota of Olive Flounder (Paralichthys olivaceus)"

_animals, 2022, doi:10.3390/ani12162043_

Round 1

Reviewer 1 Report

Table 3, HIS (%)  should be replaced by HSI (%).

Reviewer 2 Report

The present study evaluated the use of several protein sources replacing fishmeal in olive flounder diets. The authors' improvement significantly the manuscript. 

Only two minor corrections were suggested:

Lines 106-107: present levels of fishmeal protein replacement by other protein sources

Table 1 - Include some information on feed chemical composition.

Please, see attached file for more details.

Author Response

This manuscript is a resubmission of an earlier submission. The following is a list of the peer review reports and author responses from that submission.

Round 1

Reviewer 1 Report

This study looked at partial replacement of fishmeal by BSF meal and oil in diets for olive flounder. The methodology was sound, and it is a strength of the study that it considered fish nearing market size (final weight >1 kg). There are a few things that the authors should consider that will make their report stronger.

- The authors should provide details about how the experimental diets were produced. What kind of equipment was used to make pellets? Temperature of pelleting and drying, if available? 

- What were the sources of the main ingredients; BSF meal and oil, FMs, etc.

- What were the compositions of the mineral and vitamin mixes: What form of vitamin C and vitamin E were used?

- The authors should provide information about how the diets were presented to the fish. Was it by hand? Was the feed amount at a fixed ration or to satiation? Was there any accounting for uneaten diet?

- The authors should provide more details about sampling. How long before sampling were the diets withheld? Line 402 indicates that different fasting times were used. These times should be provided.

- The authors should provide survival data of the animals.

- What software was used for the analysis of the data generated?

- I don't see the value of presenting the data in graphical form in figures 1, 2, 3 and 9. Graphical representations are helpful when detecting patterns with multiple graded levels, especially when there are distinct patterns to be seen. But in this case, there are very few meaningful patterns. This comment is more of a stylistic preference. 

- The dotted line in Figure 2 doesn't refer to anything and is unneeded and misleading.

- The conclusion that BSF meal has potential should include that this study examined this ingredient with significant supplementation of essential nutrients, namely, Met, Tau, phospholipid. 

Reviewer 2 Report

Manuscript ID: animals-1779268

Major comments/suggestions

#1 Significance of this work

The authors have already published similar results (ref. No. 16) using smaller fish and diets with the formulation similar to this work.  In addition, it is well known that utilization of alternative ingredients increases as fish grow.  Thus, I am quite uncertain why the authors had to do this work, the results of which, including the examination on IGF-I and CCK, could be easily imagined.

#2 The experimental design and discussion

L77-L78, L139, L419-L484

The authors should realize that digestion in the intestine is first promoted by the enzymes secreted from the pancreas or pancreatic tissue into duodenum.  The authors examined, using intestinal tissue, the final step of digestion on the brush boarder membranes.  In this regard, the authors should have examined the activity of the pancreas and/or intestinal digesta, as the authors of ref. No. 39 did..  In addition, “fasting” could not be a topic for discussion in this work as I suggested elsewhere.

Dietary formulation, L419-L484, Table 1

The authors should realize that an equal amount of soybean meal (12%) was included in all diets including the fishmeal-based control diet.  In addition, the fishmeal was replaced in the test diets by appreciable amounts of animal ingredients (tankage meal, PBM, TBM, and BSF in FM35B) and small amounts of SPC and wheat gluten both containing less antinutritional compounds than soybean meal.  Thus, it is not useful to compare the results of this work to the previous ones using diets with fishmeal being replaced by plant ingredient(s) alone.

L349-L378

Anyway, since the inclusion level of black soldier fly meal is very small in this work, the discussion on black soldier fly here is not meaningful.  In addition, the authors should bear in mind that although the differences were not significant, the growth of flounder decreased by the inclusion of black soldier fly in both the previous and current studies.

L379-L418

The effect of “fasting” was not examined in this work and thus could not be a topic to be discussed.  In addition, the statement, “protein substitution with black…on growth” (L417-L418) could not be supported by the results.

L90, not “smaller” but probably “larger” is correct.

L95-L103, delete these duplicated presentations.

L114, add the procedure for pelleting.

L127-L128, provide the duration of food deprivation before sampling.

L197-L198, I cannot understand, “Plasma samples were used…”

Table 3, Need to add food conversion ratio (FCR) or feed efficiency ratio (FER), and feed consumption data.  The authors sometimes argued feed efficiency of previous results in Discussion.

L285, The authors merely examined goblet cells and did not examine “digestive activation factors”.

L309, L310, Is the use of the word “activity” correct?

L388, “gilthead” sea bream

L376, probably olive flounder “feed” instead of “fishmeal” is correct.

L432, not “absorption and digestibility” but “digestion and absorption” is better.

L432-L438, delete since these findings have no relevance to this work. 

L503, Did “Firmicutes” originate from black soldier fry meal per se?  In this kind of study, the microbiome in the diet should be examined.

Table 1, “Tankage meal” does not seem to be a common name.  Is this porcine blood meal?

Reviewer 3 Report

The manuscript reports the results of diets with different levels of fishmeal for olive flounder reared under commercial conditions. 

The manuscript presents a large number of parameters evaluated, most of which are adequately presented.  

Only two main points need improvement:

1) Authors need to justify the great variability of ingredients between the diets. The formulation of feeds does not allow the asset that FM levels were the only factor that affected the parameters evaluated. This variability of diets needs to be discussed in the manuscript.

2) The statistic analysis performed for the parameters evaluated over time is not correct precisely because it does not evaluate the effect of time on the results or the interaction between time and diets. 

Other minor comments were made in the attached file.

Reviewer 4 Report

The MS submitted by Seo and colleagues deals with FM reduction in olive flounders diet and reports the results in farm conditions. This is not enough to assess old experimental layouts and to pubblish a paper on Animals. 

Moreover, the MS presents several, too much, gaps in all the sections.

In the introduction section, despite arguing fish physiological mechanisms related to igf, cck, pp, etc, the authors shoud show the state of the art on olive flounders dietary formulation, on not conventional ingredients, emphasizing why the study presented could be considered of interest for readers. 

In methods, 95-111 lines are "repetita" and should be joined, as well 2.5.1 and 2.5.2 sections. In 2.5.3 section what the authors mean for "tertiary antibody"? It makes no sense. In 2.8 section the authors should deal with statistical methods, while microscopy should be included in 2.5 section.

In Results, a good histology ( as presented) is reduntant if does not show differences among the dietary groups. It should not be a morphological characterization. Histochemistry reactions and pictures are of low quality and, as presented, as well, reduntant if not showing differences. 

Discussion should be shortened and should not be a collection of  sometimes inappropriate references. 

I hope these strict comments do not discourage the authors and could be of help for future submissions.